# Diagnostic, Prognostic, and Therapeutic Value of Non-Coding RNA Expression Profiles in Renal Transplantation

**DOI:** 10.3390/diagnostics10020060

**Published:** 2020-01-22

**Authors:** Adriana Franco-Acevedo, Zesergio Melo, Raquel Echavarria

**Affiliations:** 1Pharmacology Ph.D. program, Universidad de Guadalajara, Guadalajara 44340, Mexico; ady_francoa@hotmail.com; 2CONACyT-Centro de Investigacion Biomedica de Occidente, Instituto Mexicano del Seguro Social, Guadalajara 44340, Mexico; zcmelo@conacyt.mx

**Keywords:** Non-coding RNA, renal transplantation, molecular signatures, rejection

## Abstract

End-stage renal disease is a public health problem responsible for millions of deaths worldwide each year. Although transplantation is the preferred treatment for patients in need of renal replacement therapy, long-term allograft survival remains challenging. Advances in high-throughput methods for large-scale molecular data generation and computational analysis are promising to overcome the current limitations posed by conventional diagnostic and disease classifications post-transplantation. Non-coding RNAs (ncRNAs) are RNA molecules that, despite lacking protein-coding potential, are essential in the regulation of epigenetic, transcriptional, and post-translational mechanisms involved in both health and disease. A large body of evidence suggests that ncRNAs can act as biomarkers of renal injury and graft loss after transplantation. Hence, the focus of this review is to discuss the existing molecular signatures of non-coding transcripts and their value to improve diagnosis, predict the risk of rejection, and guide therapeutic choices post-transplantation.

## 1. Introduction

End-stage renal disease (ESRD) is an increasingly prevalent public health problem responsible for millions of deaths worldwide each year [1,2]. Transplantation is considered the best alternative for patients with ESRD as it reduces complications, increases survival, and improves the quality of life [3]. Despite continuous improvements in pre-transplant donor-recipient screening, immunosuppressive treatments, and overall management post-transplantation, rejection remains a challenge for long-term allograft survival [4].

Non-coding RNAs (ncRNAs) are RNA molecules that lack protein-coding potential but are essential in the regulation of epigenetic, transcriptional, and post-translational mechanisms involved in homeostasis and disease [5]. The development of RNA sequencing (RNA-seq) technologies has revealed that ncRNAs in eukaryotic transcriptomes are not only highly abundant but also incredibly diverse [6]. Species of ncRNAs include small nuclear RNAs, small nucleolar RNAs, microRNAs (miRNAs), Piwi-interacting RNAs, circular RNAs (circRNAs), and long ncRNAs (lncRNAs). Nephrological studies on ncRNA regulation and function, though scant, have advanced our knowledge of disease initiation and progression [7,8,9]. Furthermore, the ability of miRNAs, lncRNAs, and circRNAs to influence biological pathways dysregulated in disease, in addition to their stability in tissue and biofluids, make them excellent candidates for biomarker discovery [10]. 

MiRNAs are small RNA molecules of ~23 nucleotides in length that interact with the 3′ untranslated region (UTR) of their target mRNAs to induce their degradation or repress their translation [11]. Nonetheless, miRNAs can also interact with 5′ UTRs and coding sequences, as well as play distinct roles as negative gene transcription regulators or positive post-transcriptional modulators of gene expression [12,13]. Renal cortex expression of miRNAs involved in organ development and homeostasis has been documented [14]. 

CircRNAs expressed in mammalian cells are non-coding transcripts with a circular structure in which the 3′ and 5′ ends are covalently linked [15]. These ncRNAs arise from direct back-splicing, a mechanism in which an exon at the 3′ end of a gene is back-spliced to an exon at the 5′ end of the gene resulting in a circular molecule [16]. CircRNAs are abundant and highly stable because their structure allows them to avoid the degradative action of exonucleases [15,17]. The cellular functions of circRNAs are beginning to emerge. CircRNAs located mainly in the nucleus appear to have a regulatory function in transcription, whereas cytoplasmic circRNAs participate in post-transcriptional gene regulation [18,19,20]. CircRNAs also act as miRNA sponges and possess scaffolding functions that affect protein–protein interactions [21,22]. Although the expression and functional role of circRNAs in the kidney remains mostly unknown, some studies have postulated circRNAs as potential therapeutic targets in renal diseases [23,24].

lncRNAs are transcripts of > 200 nucleotides in length, with no protein-coding capacity and often transcribed from intergenic regions [5,25]. These lncRNAs maintain cellular identity and homeostasis through the regulation of gene expression at transcriptional and post-transcriptional levels, chromatin modulation, competing endogenous RNAs, and protein scaffolding [26,27]. Moreover, lncRNAs can interact with epigenetic regulators and transcription factors in a sequence-independent manner through non-coding transcription [5]. Multiple studies link lncRNA dysregulation and pathological disease states that affect the kidney, such as acute kidney injury (AKI), chronic kidney disease, nephropathies, and allograft rejection [27,28,29].

The purpose of this review is to summarize current evidence regarding the value of ncRNA expression profiles in renal transplantation. All the information gathered in this integrative review was extracted from original and high-quality English language papers indexed in PubMed, based on the search of the MeSH Terms: “non-coding RNA”, “microRNA”, “long non-coding RNA”, “circular RNA”, “renal transplantation”, “molecular signatures”, and “allograft rejection”. 

## 2. Molecular Profiling in Nephrology and Transplantation

Extraordinary progress in technologies designed for high-throughput data generation and analysis has had profound repercussions on biomedical research and precision medicine [30]. Unfortunately, nephrology has stayed behind this big-data revolution compared to other medical fields, as evaluation of renal function and clinical decision-making is still carried out by outdated methods [30]. Furthermore, the pace at which we are acquiring the knowledge necessary to achieve a timely and more accurate diagnosis of kidney disease and allograft rejection post-transplantation is languid compared to the accelerated epidemic associated with ESRD [1].

Renal biopsy is the gold standard in current clinical practice for the diagnosis of post-transplant rejection. Nevertheless, the data is descriptive and provides little information about prognosis [31]. The field of transcriptomics has emerged to provide a more comprehensive characterization of gene expression in renal biopsies of kidney transplant recipients [32,33,34,35]. Moreover, studies in circulating blood cells and urine sediment provide evidence to support non-invasive transcriptional analysis as a viable alternative to the biopsy procedure for their ability to deliver predictive, diagnostic, and prognostic data [36,37]. 

Molecular signatures of ncRNAs can echo epigenetic and post-transcriptional modifications occurring within the kidney [30,38,39]. Thus, the study of non-coding transcriptomes in transplantation could prove useful not only to understand disease-associated renal phenotypes, but to establish novel therapeutic targets, and develop more sensitive and specific diagnostic procedures. A summary of ncRNAs (miRNAs, lncRNAs, and circRNAs) that have been associated with disease states post-transplantation in renal recipients is described in Table 1.

## 3. Non-Coding RNA Profiles as Predictors of Renal Phenotypes

### 3.1. Post-Transplant AKI

Delayed graft function (DGF) is the clinical manifestation of post-transplant AKI, a common complication that affects short- and long-term transplantation outcomes [66]. Organs procured from deceased and expanded criteria donors have more extensive ischemic damage that leads to a higher incidence of DGF [67,68]. Thus, avoiding DGF is clinically relevant in the context of a limited donor pool and the increasing use of expanded criteria donor kidneys. 

Wilflingseder J. et al. initially identified seven miRNAs (miR-182-5p, miR-21-3p, miR-106a/b, miR-20a, miR-18a, and miR-17) upregulated in DGF kidneys [40]. Then, the same group subjected zero-hour and follow-up biopsies to genome-wide mRNA-miRNA profiling and further validated two miRNAs (miR-182-5p and miR-21-3p) as strongly associated with post-transplant AKI and DGF [41]. MiR-182-5p is a post-transcriptional regulator of genes involved in apoptosis, cell-cycle regulation, T-cell differentiation, and migration. Renal inhibition of miR-182-5p in vivo by an antisense oligonucleotide improved kidney function and morphology after AKI, confirming the role of this miRNA in the pathogenesis of ischemic injury [69]. Hypoxia induces miR-21, and its expression contributes to glomerular and tubulointerstitial pathogenesis and renal fibrosis in AKI, IgA nephropathy, and diabetic nephropathy [70]. In addition to miR-182-5p and miR-21, the upregulation of miR-146a-5p expression in biopsy samples discriminates between patients with DGF versus acute rejection (AR) and stable patients [42]. MiR-146a-5p downregulates the nuclear factor-kappa B (NF-κB signaling pathway, exhibits a protective role against hypoxia-induced apoptosis and inflammation, and is considered a potential serum biomarker of non-transplant AKI [71,72,73]. 

Interestingly, cell-free miRNAs in graft preservation fluid are predictive of DGF post-transplantation. Roest HP et al. identified an association between high levels of miR-505-3p in the preservation fluid of kidney grafts donated after circulatory death and an increased risk of DGF after transplantation [43]. Another prospective cohort study that analyzed graft dysfunction in transplant recipients of expanded criteria donor organs confirmed the significance of a subset of four miRNAs (miR-486-5p, miR-144-3p, miR-142-5p, and miR-144-5p) previously identified in DGF development [44]. This same group also studied miRNAs in the preservation solutions from human allografts and found an expression signature of eleven miRNAs (miR-486, miR-18a, miR-20a, miR-363-3p, miR-144-3p, miR-454-3p, miR-223-3p, miR-142-5p, miR-502-3p, miR-144-3p, and miR-144-5p) that could predict post-transplant DGF [74]. Moreover, three miRNAs (miR130a-3p, miR-30e-5p and miR-324-3p) were associated with a decrease in renal function one year post-transplantation. These studies highlight the value of molecular ncRNA signatures in perfusion fluid as a source for the identification of organ viability biomarkers and the development of post-transplant AKI. 

Exosomes are secreted vesicles present in biological fluids that transfer molecular cargo, including ncRNAs, between cells. Despite current excitement regarding exosomes as up-and-coming diagnostic and prognostic tools in cancer, studies of exosomes and other extracellular vesicles in transplantation are scarce [75]. Wang J. et al. found an exosome-derived miRNA signature associated with DGF using high-throughput sequencing [45]. Exosomes were isolated from peripheral blood of renal transplant recipients that presented with DGF one week post-transplantation, followed by the analysis of their miRNA cargo. The result showed a differential expression of fifty-seven exososome-derived miRNAs mainly expressed in kidney recipients with DGF. Moreover, miR-33a-5p and miR-151a-5p were significantly upregulated in DGF and correlated with creatinine and blood urea nitrogen levels within the first week after transplantation. The authors suggest that exosomal expression of miR-33a-5p and miR-151a-5p, together with miR-98-5p, could be an alternative for new biomarkers in the detection of DGF. 

Although it is clear that other types of ncRNAs such as lncRNA and circRNAs are involved in the development of renal ischemic injury in animal models, their role as potential predictors of DGF in transplant recipients is unknown.

### 3.2. Allograft Rejection

Traditional diagnosis of allograft pathology post-transplantation relies heavily on the histological assessment of renal biopsies following the Banff classification [76]. The possibility of using archived histopathological tissues for molecular RNA profiling has encouraged the search of molecular signatures able to discriminate between disease states while also providing insight into disease pathophysiology [77]. The purpose of this section is to describe some of the studies that have identified ncRNA signatures, miRNAs, and lncRNAs able to distinguish between distinct renal phenotypes associated with allograft rejection. 

One of the first microarray analysis of ncRNAs in renal biopsies led to the identification of twenty differentially expressed miRNAs in AR biopsies [46]. Sui W. et al. found twelve upregulated miRNAs in AR biopsies (miR-324-3p, miR-611, miR-654, miR-330, miR-524*, miR-17-3p, miR-483, miR-663, miR-516-5p, miR-326, miR-197, and miR-346), as well as eight downregulated miRNAs (miR-658, miR-125a, miR-320, miR-381, miR-628, miR-602, miR-629, and miR-125a). Anglicheau D. et al. also investigated whether miRNA expression profiles in kidney biopsies are diagnostic of AR and can predict allograft function [47]. Their results demonstrate that intragraft levels of six miRNAs (miR-142–5p, miR-155, miR-223, miR-10b, miR-30a-3p, and let-7c) are diagnostic of AR. The miRNAs miR-142–5p, miR-155, and miR-223 each predicted AR with > 90% sensitivity and specificity. Strikingly, this miRNA signature was superior to intragraft levels of mRNAs associated with AR (CD3, CD20, NKCC2, and USAG1), which, though also diagnostic of AR, showed less combined sensitivity and specificity. Immune cell infiltration, particularly of T- and B-cells, is characteristic of AR, and various miRNAs found dysregulated in AR biopsies are expressed in hematopoietic cells and have been linked to innate and adaptive immune processes [48,76,78,79,80]. There is evidence that a profile of four miRNAs (miR-142-5p, miR-142-3p, miR-155, and miR-223), specific for hematopoietic lineage, can discriminate acute T-cell-mediated rejection (TCMR) from normal allografts [81]. 

An analysis that integrated transcription factors, miRNAs, and lncRNAs resulted in a comprehensive view of signaling pathways in AR [49]. Through protein array-based proteomics and RNA microarray-based genomics of renal biopsies, five transcription factors (AP-1, AP-4, STATx, c-Myc, and p53), twelve miRNAs (miR-125b, miR-483, miR-663, miR-326, miR-346, miR-125a, miR-381, miR-602, miR-629, miR-324-3p, miR-658, and miR-524*), and thirty-two lncRNAs (NR_026695, NR_024080, uc010kwo, NR_023318, NR_026576, NR_002909, NR_026550, NR_003024, NR_027303, NR_003130, NR_024418, uc003wcs, uc003syy, uc002zic, uc010gqe, uc003bgk, uc003akf, NR_024400, NR_024332, uc001pyd, uc010lqx, NR_024611, uc002zpx, NR_001562, NR_002941, uc010akv, uc002nyb, NR_003573, NR_002791, uc003zfx, uc003dwf, and uc003tsq) associated with AR were identified. A subsequent study analyzed ncRNA expression using lncRNA arrays in renal allograft biopsies of AR and found more than 5000 altered lncRNAs [39]. Pathway analysis revealed that the top significantly enriched pathways for the differentially expressed lncRNAs were immune-related and included interleukin 2-mediated signaling, interleukin 6, B-cell survival, and NF-κB activation. Additionally, Chen W et al. validated five upregulated lncRNAs (uc001fty, uc003wbj, AKI129917, uc010ftb, and AF113674) as potential biomarker candidates. These lncRNAs participate in host-defense responses, inflammatory signaling, antigen-presentation, apoptosis, and complement activation [39,82,83]. However, their function in allograft rejection is still unknown. 

A few studies have addressed the role of ncRNAs associated with transforming growth factor-beta (TGF-β), a key promoter of fibrosis and renal failure [50]. Acute pyelonephritis exhibits overlapping histologic features with AR. Thus, Oghumu S et al. searched for miRNA profiles to distinguish between allograft acute pyelonephritis and AR. They validated five miRNAs (miR-99b, miR-23b, let-7b-5p, miR-30a, and miR-145) from a panel of twenty-five miRNAs whose expression was statistically different between acute pyelonephritis and AR [51]. TGF-β has been associated with susceptibility to recurrent urinary tract infections in humans [52]. Interestingly, some of the miRNAs from the acute pyelonephritis signature modulate TGF-β signaling pathways, which could hint to their role in fibrosis and renal scarring progression [61,62,84,85,86]. Meanwhile, Qiu J et al. studied the expression of the lncRNA-activated by TGF-β (lncRNA-ATB) in AR and its role in the nephrotoxicity induced by the immunosuppressive drug Cyclosporin A [60]. Renal biopsies of AR have higher expression of LncRNA-ATB, which is inversely correlated with miR-200c expression, suggesting that LncRNA-ATB acts as a competing endogenous RNA [87]. LncRNA-ATB has three potential binding sites for the miR-200 family, but the strongest association occurs between lncRNA-ATB and miR-200c [88]. The lncRNA-ATB promotes epithelial–mesenchymal transition by acting as a sponge for miR-200c, and TGF-β1 plays a role in this mechanism [89]. Lnc-ATB is activated by TGF-β1 and high levels of miR-200c reverse TGF-β1 actions. Additionally, ectopic expression of lncRNA-ATB in renal tubular epithelial cells potentiates Cyclosporine A-mediated apoptosis [60].

Few research groups have associated ncRNA signatures in kidney biopsies with chronic allograft dysfunction (CAD) and antibody-mediated rejection (ABMR). A study by Scian MJ et al. used microarrays to find a miRNA signature of CAD with interstitial fibrosis and tubular atrophy (IF/TA) [51]. Differential expression of five miRNAs (miR-142-3p, miR-204, miR-107, miR-211, and miR-32) was detected in these biopsies. Moreover, this miRNA signature could discriminate patients with consistent IF/TA-like miRNA changes that correlated with histological findings even when their serum creatinine and estimated glomerular filtration rate measurements indicated normal renal function. Heinemann FM et al. proposed a miRNA signature associated with human leukocyte antigens (HLA) class I-donor specific antibodies (DSA) in microdissected glomeruli of 20 human transplant biopsies after studying a set of sixteen candidate miRNAs initially discovered in a glomeruloendothelial in vitro model of ABMR [52]. Ten miRNAs were upregulated (let-7c-5p, miR-28-3p, miR-30d-5p, miR-99b-5p, miR-125a-5p, miR-195-5p, miR-374b-3p, miR-484, miR-501-3p, and miR-520e) and two downregulated (miR29b-3p and miR-885-5p) in DSA-positive transplant recipients, compared to matched controls without ABMR.

Integrating information from curated transcriptomic datasets stored in database repositories of high-throughput gene expression data, hybridization arrays, chips, and microarrays such as Gene Expression Omnibus (GEO, www.ncbi.nlm.nih.gov/geo/) is an excellent opportunity to circumvent a critical limitation of ncRNA profiling studies in kidney biopsies carried out to date, which is the small number of patients included in each study. Using this approach, Zou Y et al. performed lncRNA data mining in 1105 human renal allograft biopsies of AR in GEO, and discovered six highly expressed lncRNAs (RP11-25K19.1, ITGB2- AS1, MIR155HG, CARD8-AS1, RP6-159A1.4, and TRG-AS1) strongly associated with AR [61]. A three lncRNA signature (ITGB2-AS1, MIR155HG, and CARD8-AS1) allowed the generation of a risk score predictive of graft loss (AUC = 0.73). Moreover, the lncRNA MIR155HG, the host gene of miR-155, was associated with AR, risk of graft loss, and TCMR [61,84]. Unfortunately, this study was unable to determine if lncRNA expression is an independent predictor of graft loss because clinical information was absent from the GEO datasets. Similarly, Xu J et al. identified lncRNAs associated with chronic damage and graft loss post-transplantation using the GEO datasets of 407 biopsies from three different studies [62]. Chronic damage, progressive histological damage, and graft failure significantly associated with high expression of AC093673.5 across all datasets and a signature of six lncRNAs (AC126763.1, RP11-280K24.1, LINC01137, WASIR2, RP1-276N6.2, and AD000684.2) could predict both development and progression of CAD. 

## 4. Circulating Non-Coding RNAs in Transplantation

There is a link between altered concentrations of ncRNAs released into the circulation by organs and tissues and various disease states, including those affecting the kidneys [7,9]. Since ncRNAs are stable and easily detectable in body fluids, as well as expressed in circulating blood cells, they are suitable for diagnostic and prognostic applications [10,85]. Circulating ncRNA signatures are promising, non-invasive biomarkers of diseases that affect the kidney [86]. In this section, we highlight some of the most important circulating ncRNAs that have emerged from transcriptomic studies. Their diagnostic, prognostic, and therapeutic value in transplantation is summarized below and in Figure 1.

### 4.1. MicroRNAs

MiR-99a belongs to the let-7c/miR-99a/miR-125b cluster (21q21.1), targets the AKT/mTOR signaling pathway, and is involved in inflammation, cell migration, and proliferation [90,91,92,93]. The expression of miR-99a is dysregulated in plasma, peripheral blood mononuclear cell (PBMCs), and urine of patients with AR, CAD with IF/TA, and TCMR with discrepancies among studies [47,53,54].

MiR-21 is a hypoxia-controlled miRNA consistently upregulated in kidney diseases, promotes renal fibrosis via post-transcriptional silencing of metabolic pathways, and is a known mediator of inflammatory responses associated with NF-κB activation and cytokine expression [94,95,96]. The expression of miR-21 was found upregulated in biopsies, urine, and PBMCs from CAD patients [55,56].

MiR-155 is considered a master regulator of inflammation and participates in the differentiation and stimulation of cells related to innate and adaptive immune responses [97,98]. The miRNA miR-155 is upregulated in PBMCs, as well as in urine of AR and CAD patients [47,48,99]. MiR-155 expression decreases after an AR episode, when immunosuppressive therapies are successful. Thus, it is considered a prognostic and predictive urinary biomarker of rejection, transplantation outcome, and treatment response [99].

MiR-142-3p is found in hematopoietic cells and has been implicated in the control of immune functions, particularly regulatory T-cells, by targeting the transcription factor forkhead box P3 and autophagy [100]. MiR-142-3p is upregulated in PBMC, plasma, and urine samples of CAD with IF/TA [48,57]. Janszky N. et al. mentioned that combined quantifications of miR-142-3p in urine, blood, PBMC, or biopsy could be an option for accurate diagnostic in AR or CAD, for its potential to detect inflammatory graft injury after kidney transplantation [57]. In contrast, miR-142-5p expression in PBMCs efficiently discriminates chronic ABMR from stable graft function and renal failure [101]. Additionally, miR-142-5p expression was not influenced by immunosuppressive treatment, and its predicted mRNA targets were downregulated in a microarray data-set of PBMCs in patients diagnosed with ABMR.

MiR-223, a miRNA expressed in immune cells, is downregulated in patients with chronic kidney disease and upregulated in PBMCs of transplant recipients with experiencing TCMR [48,58]. The sensitivity and specificity of miR-223 as a biomarker of acute TCMR have been reported as high.

MiR-210, along with miR-10a and miR-10b, is downregulated in the urine of AR patients [59]. Remarkably, miR-210 presents sexual dimorphism, and its expression is higher in women. MiR-210 can diagnose AR, is associated with a decline in glomerular filtration rates one year after transplantation, and is responsive to successful immunosuppressive therapy. MiR-210 expression is induced in low oxygen concentration by the hypoxia-inducible factor to reduce ischemic damage to the kidney, which could partly explain its modulation in AR [102].

### 4.2. Long Non-Coding RNAs

RP11-354P17.15-001, a lncRNA induced by inflammatory signaling in tubular epithelial cells, is upregulated in urine samples from patients with acute TCMR. This lncRNA detects acute TCMR with high specificity, has a positive association with a decline in glomerular filtration rate six weeks post-transplant, and its expression normalizes in AR patients with effective anti-rejection therapy [63].

AF264622 and AB209021 are two lncRNAs upregulated in peripheral blood from pediatric and adult renal transplant patients with AR. Their high sensitivity and specificity to detect AR make them a promising alternative in the search for non-invasive biomarkers of AR in both pediatric and adult renal transplant recipients [28].

MGAT3-AS1 is an intronic antisense lncRNA downregulated in mononuclear cells from deceased donor kidney transplant recipients with DGF [64]. Thus, MGAT3-AS1 is a potential biomarker of short-term transplantation outcomes in patients with deceased donor kidneys at risk of DGF.

### 4.3. Circular RNAs

Regarding hsa_circ_0001334 and hsa_circ_0071475, the urinary expression of these circRNAs was significantly and specifically increased in patients with acute TCMR when compared to stable grafts without signs of rejection [65]. This finding was made by Kölling M et al., who performed a genome-wide analysis of circRNA expression in the urine of transplant patients with acute TCMR. A total of 5119 circRNAs were detected, out of which a distinct expression signature of twenty circRNAs could discriminate AR patients from patients with stable grafts without signs of rejection. The expression of circRNA hsa_circ_0001334 was associated with a decline in glomerular filtration rate six weeks after transplantation. Moreover, hsa_circ_0001334 expression detects patients with subclinical rejection, and its urinary levels normalized in patients with acute TCMR that respond successfully to anti-rejection therapy. Hence, hsa_circ_0001334 could be a biomarker of AR potentially superior to serum creatinine.

## 5. Conclusions

The generation and validation of dependable and robust ncRNA expression profiles able to improve clinical and pathological factors currently being used to assess disease states and response to treatment is a promising alternative to prolong graft survival in renal transplant recipients. Unfortunately, studies addressing the diagnostic, prognostic, and therapeutic value of ncRNAs in renal transplantation to date are insufficient. Future research must include more massive data-sets and address the need for large-scale RNA-seq studies. Molecular profiling in nephrology and transplantation faces multiple challenges. First, transplantation outcomes are affected by multiple factors, and disease phenotypes post-transplantation are complex. Second, the degree of certainty in the diagnosis of allograft rejection through the subjective assessment of histopathological lesions is inadequate. Some biopsies do not fit within current histological classifications and end up being classified as borderline or indeterminate rejections. Third, biopsies are invasive and can present complications, which makes it hard to have timely detection of disease states post-transplantation. Studying patterns of variation of ncRNA expression post-transplantation could be a step forward into personalized treatments and better diagnostic methods that could help us overcome the actual constraints imposed by existing histologic classes.

## Figures and Tables

**Figure 1 diagnostics-10-00060-f001:**
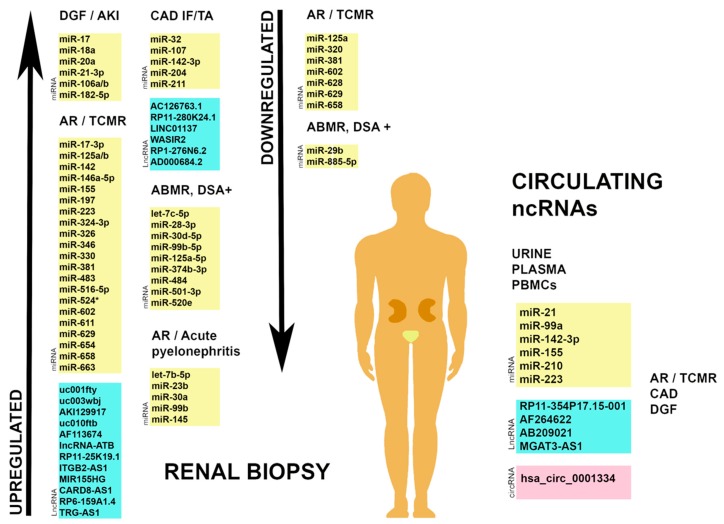
Renal and circulating non-coding RNAs in transplantation. Multiple studies have described molecular signatures of ncRNAs, as well as individual miRNAs, lncRNAs and circRNAs associated with disease states post-transplantation. Their study could lead to improved diagnosis of transplantation outcomes while simultaneously offering an insight into the pathogenesis of renal disease often developed in these patients. miRNA, microRNA; lncRNA, long non-coding RNA; circRNA, circular RNA; DGF, delayed graft function; AKI, acute kidney injury; AR, acute rejection; TCMR, T-cell-mediated rejection; CAD, chronic allograft dysfunction; IF/TA, interstitial fibrosis and tubular atrophy; ABMR, antibody-mediated rejection; DSA, donor specific antibodies.

**Table 1 diagnostics-10-00060-t001:** Expression of ncRNAs (non-coding RNAs) associated with disease states post-transplantation.

Authors	Non-Coding RNAs	Expression	Localization	Disease State
**miRNAs**				
Wilflingseder, J. et al. [40,41]	miR-182-5p, miR-21-3p, miR-106a/b, miR-20a, miR-18a, miR-17	Upregulated	Biopsy	DGF, AKI
Milhoransa, P. et al. [42]	miR-146a-5p	Upregulated	Biopsy	AR
Roest, H.P. et al. [43]	miR-505-3p	Upregulated	Preservation fluid	DGF
Gómez-Dos-Santos, V. et al. [44]	miR-486, miR-18a, miR-20a, miR-363-3p, miR-144-3p, miR-454-3p, miR-223-3p, miR-142-5p, miR-502-3p, miR-144-3p, miR-144-5p	Upregulated	Preservation solution	DGF
Wang, J. et al. [45]	miR-33a-5p, miR-151a-5p, miR-98-5p	Upregulated	Exosomes from peripheral blood	DGF
Sui, W. et al. [46]	miR-324-3p, miR-611, miR-654, miR-330, miR-524 *, miR-17-3p, miR-483, miR-663, miR-516-5p, miR-326, miR-197, miR-346	Upregulated	Biopsy	AR
	miR-658, miR-125a, miR-320, miR-381, miR-628, miR-602, miR-629	Downregulated		
Anglicheau, D. et al. [47]	miR-142-5p, miR-155, miR-223	Upregulated	Biopsy	AR
Soltaninejad, E. et al. [48]	miR-142-5p, miR-142-3p, miR-155, miR-223	Upregulated	Biopsy	TCMR
Sui, W. et al. [49]	miR-483, miR-381, miR-602, miR-629, miR-658, miR-524 *, miR-125a/b, miR-324-3p, miR-663, miR-326, miR-346	Varies	Biopsy	AR
Oghumu, S. et al. [50]	miR-99b, miR-23b, let-7b-5p, miR-30a, miR-145	Varies	Biopsy	AR, acute pyelonephritis
Scian, M.J. et al. [51]	miR-142-3p, miR-204, miR-107, miR-211, miR-32	Varies	Biopsy, urine	CAD IF/TA
Heinemann, F.M. et al. [52]	let-7c-5p, miR-28-3p, miR-30d-5p, miR-99b-5p, miR-125a-5p, miR-195-5p, miR-374b-3p, miR-484, miR-501-3p, miR-520e	Upregulated	Biopsy	ABMR DSA+
	miR-29b-3p, miR-885-5p	Downregulated		
Anglicheau, D. et al. [47] Tao, J. et al. [53] Ledeganck, K.J. et al. [54]	miR-99a	Varies	Plasma, PBMCs, urine	AR, CAD IF/TA, TCMR
Ben-Dov, I.Z. et al. [55] Zununi Vahed, S. et al. [56]	miR-21	Upregulated	Biopsy, urine, PBMCs	CAD
Anglicheau, D. et al. [47] Soltaninejad, E. et al. [48] Zununi Vahed, S. et al. [56]	miR-155	Upregulated	PBMCs, urine	AR, CAD, Responsive to immunossuppresive therapy
Soltaninejad, E. et al. [48] Janszky, N. et al. [57]	miR-142-3p	Upregulated	PBMCs, plasma, urine	CAD IF/TA
Soltaninejad, E. et al. [48] Ulbing, M. et al. [58]	miR-223	Varies	PBMCs	CAD TCMR
Lorenzen, J.M. et al. [59]	miR-210	Downregulated	Urine	AR
	**lncRNAs**			
Sui, W. et al. [49]	NR_026695, NR_024080, uc010kwo, NR_023318, NR_026576, NR_002909, NR_026550, NR_003024, NR_027303, NR_003130, NR_024418, uc003wcs, uc003syy, uc002zic, uc010gqe, uc003bgk, uc003akf, NR_024400, NR_024332, uc001pyd, uc010lqx, NR_024611, uc002zpx, NR_001562, NR_002941, uc010akv, uc002nyb, NR_003573, NR_002791, uc003zfx, uc003dwf, and uc003tsq	Varies	Biopsy	AR
Chen, W. et al. [39]	uc001fty, uc003wbj, AKI129917, uc010ftb, AF113674	Upregulated	Biopsy	AR
Qiu, J. et al. [60]	lncRNA-ATB	Upregulated	Biopsy	AR, Cyclosporine-induced nephrotoxicity
Zou, Y. et al. [61]	RP11-25K19.1, ITGB2- AS1, MIR155HG, CARD8-AS1, RP6-159A1.4, TRG-AS1	Upregulated	Biopsy **	AR
Xu, J. et al. [62]	AC126763.1, RP11-280K24.1, LINC01137, WASIR2, RP1-276N6.2, AD000684.2	Upregulated	Biopsy **	CAD
Lorenzen, J.M. et al. [63]	RP11-354P17.15-001	Upregulated	Urine	TCMR
Ge, Y.Z. et al. [28]	AF264622, AB209021	Upregulated	Blood	AR
Nagarajah, S. et al. [64]	MGAT3-AS1	Downregulated	PBMCs	DGF
	**circRNAs**			
Kölling, M. et al. [65]	hsa_circ_0001334, has_circ_0071475	Upregulated	Urine	TCMR

miRNA, microRNA; lncRNA, long non-coding RNA; circRNA, circular RNA; DGF, delayed graft function; AKI, acute kidney injury; *AR*, acute rejection; TCMR, T-cell-mediated rejection; CAD, chronic allograft dysfunction; IF/TA, interstitial fibrosis and tubular atrophy; ABMR, antibody-mediated rejection; DSA, donor specific antibodies. ** Analysis from GEO Datasets.

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
