# Peer review of "Diagnostic, Prognostic, and Therapeutic Value of Non-Coding RNA Expression Profiles in Renal Transplantation"

_diagnostics, 2020, doi:10.3390/diagnostics10020060_

Round 1

Reviewer 1 Report

The manuscript is well written and provides a thorough review in renal transplantation related to non-coding RNA (ncRNA). This will be of great benefit to a related area of study, especially, who is interested in ncRNA. I don’t have any major concern but have several comments for Table 1.

Please provide a description on how these articles as well as GEO datasets in Table 1 were collected/selected. That is, please specify the selection criterion(a), the name of database used (such as, PubMed, Google scholar, etc.) and corresponding keywords if used keywords, etc. Please describe these as a separate section. Please add the publication year in Table 1. Please also add the area of application (i.e., diagnostics, prognostics, therapeutic, etc.),of the ncRNA in Table 1. If applicable, it will be good if the authors specify whether this is a partial or comprehensive review

Author Response

Dear reviewer, thank you for your time and your comments. The following changes have been made to the manuscript:

Point 1: The manuscript is well written and provides a thorough review in renal transplantation related to non-coding RNA (ncRNA). This will be of great benefit to a related area of study, especially, who is interested in ncRNA. I don’t have any major concern but have several comments for Table 1. Please provide a description on how these articles as well as GEO datasets in Table 1 were collected/selected. That is, please specify the selection criterion(a), the name of database used (such as, PubMed, Google scholar, etc.) and corresponding keywords if used keywords, etc. Please describe these as a separate section. Please add the publication year in Table 1. Please also add the area of application (i.e., diagnostics, prognostics, therapeutic, etc.),of the ncRNA in Table 1. If applicable, it will be good if the authors specify whether this is a partial or comprehensive review.

Response:

A short paragraph describing how the articles included in the review were selected, database used (PubMed), and keywords was added at the end of Section 1 (Introduction). Additonal changes were made to improve Table 1, including the year of publication.

Reviewer 2 Report

In this review, the Authors focus on the role of ncRNAs both in health and in disease. Specifically they analyse their involvement in renal injury and graft loss after transplantation.

The manuscript is interesting and well written. I suggest dealing with the following implementations.

The authors describe the involvement of miRNA in the DGF, the chemical manifestation of post-transplant AKI. The discussion should be extended to other classes of ncRNAs or the putative role of these ncRNAs should be proposed/discussed. The molecular mechanism by which lncRNAs ATB could regulate miR200c expression should be addressed. The authors describe a sufficient number of ncRNAs involved in these diseases. However, when they focus on circulating miRNAs, they should analyse the role of extracellular vesicles in this context since they can be loaded with these ncRNAs.

Author Response

Dear reviewer, we appreciate your comments. The following changes have been made to the manuscript:

To address the molecular mechanism by which lncRNAs ATB could regulate miR200c expression a paragraph was added in line 186. Additionally, we added a small paragraph addressing the role of extracelular vesicles, paticularly exosomes, in this context in line 133. We found an interesting study of miRNA high-throughput profiling in exosomes of transplant recipients with DGF by Wang J. et al. (2019) which complemented the rest of the studies mentioned in this review. A small paragraph refering to the role of other ncRNAs in DGF and how much remains unknwon was added in line 145.